# Non-Steroidal Estrogens Inhibit Influenza Virus by Interacting with Hemagglutinin and Preventing Viral Fusion

**DOI:** 10.3390/ijms242015382

**Published:** 2023-10-19

**Authors:** Elisa Franzi, Gregory Mathez, Soraya Dinant, Charlotte Deloizy, Laurent Kaiser, Caroline Tapparel, Ronan Le Goffic, Valeria Cagno

**Affiliations:** 1Institute of Microbiology, Lausanne University Hospital, University of Lausanne, 1011 Lausanne, Switzerland; 2INRAE, UVSQ, UMR892 VIM, Université Paris-Saclay, 78350 Jouy-en-Josas, France; 3Laboratory of Virology, Division of Infectious Diseases and Division of Laboratory Medicine, University Hospitals of Geneva, University of Geneva, 1206 Geneva, Switzerland; 4Center for Emerging Viruses, Geneva University Hospitals, 1205 Geneva, Switzerland; 5Department of Microbiology and Molecular Medicine, University of Geneva, 1206 Geneva, Switzerland

**Keywords:** influenza virus, hemagglutinin, entry, screening, antiviral, estrogens

## Abstract

Influenza virus is one of the main causes of respiratory infections worldwide. Despite the availability of seasonal vaccines and antivirals, influenza virus infections cause an important health and economic burden. Therefore, the need to identify alternative antiviral strategies persists. In this study, we identified non-steroidal estrogens as potent inhibitors of influenza virus due to their interaction with the hemagglutinin protein, preventing viral entry. This activity is maintained in vitro, ex vivo, and in vivo. Therefore, we found a new domain to target on the hemagglutinin and a class of compounds that could be further optimized for influenza treatment.

## 1. Introduction

Influenza virus is one of the leading causes of respiratory infections. The effects of the virus can range from mild respiratory symptoms to death, particularly in vulnerable populations such as the elderly and the immunocompromised [1]. It is estimated that influenza virus alone causes up to 650,000 deaths per year [2]. Furthermore, influenza A viruses have the potential to cause pandemics due to zoonotic spillovers, which could cause more severe symptoms due to the absence of previous immunity [1]. This zoonotic threat is of particular concern due to the current epizootic of the highly pathogenic avian H5N1 virus. New outbreaks are occurring almost daily on an unprecedented scale, and numerous transmissions to mammals have been described, raising fears that the virus is adapting to infect humans [3].

Luckily, seasonal vaccines and antivirals are available; however, both are far from ideal. Vaccines are based on predictions of the circulating strains and offer suboptimal protection [4], while the different classes of antivirals present limitations, such as their limited window of administration and the risk of selecting for resistance [1].

Inhibitors of the proton pump M2, such as adamantanes, are no longer used since circulating strains of influenza A virus are not sensitive to it [5]. The first line of antivirals in use are inhibitors of neuraminidase (oseltamivir, zanamivir, and peramivir). They are beneficial if administered within 48 h of symptom onset, and resistance has been documented for all available options [6]. For instance, before the advent of the H1N1 pandemic strain of 2009, the majority of H1N1 strains were resistant to oseltamivir [1], showing that resistant strains are fit to circulate widely. Finally, the most recent class of inhibitors targets the subunit of the polymerase PA [7]. However, baloxavir marboxil presents similar features to neuraminidase inhibitors, with a limited administration window and the possibility of selecting resistant viruses [8].

For these reasons, it is important to identify new antivirals effective against influenza with complementary mechanisms of action in order to enlarge the arsenal of weapons active against influenza and develop potential combinatorial therapies.

In order to shorten the time to the clinic, libraries of existing compounds can be screened to find a second use [9]. This strategy has been largely applied to viruses, particularly with SARS-CoV-2 [10,11], with the identification of small molecules showing activity in vitro and in vivo. 

For these reasons, we screened a library composed of FDA-approved compounds against influenza A virus. One of the top hits of the screening was diethylstilbestrol (DES), an estrogen active against multiple strains of influenza virus in cell lines and human-derived respiratory tissues. The compound interacts with hemagglutinin, preventing viral entry. Structurally related compounds show similar inhibitory activity, and the best option in terms of the selectivity index, hexestrol, showed a protective effect in mice infected with influenza A virus.

## 2. Results

### 2.1. DES Is the Top Hit of a Screening of the Prestwick Library against Influenza A Virus

The Prestwick FDA-approved drug library was screened against H1N1 A/Netherlands/602/2009 (N09) in A549 cells by adding the compounds 1 h before and during infection. The results were analyzed by immunostaining and by viability measurement, and by matching the two analyses, a list of the top five hits was obtained. The results are shown in Table 1 and include the relative percentage of inhibition with immunostaining and the percentage of increase in viability if compared to infected wells. DES was subjected to further analysis since no reports of the antiviral activity of this molecule were available.

### 2.2. Dienestol, Hexestrol, and Bakuchiol Show Comparable Antiviral Activity

To verify which structural features are important for the antiviral activity of DES, we tested analogs of DES (dienestrol, hexestrol, masoprocol, bakuchiol, and resveratrol), selected using the software Swiss Similarity (http://www.swisssimilarity.ch/) [12], for their effectiveness against N09. The molecules were added to MDCK cells 1 h before infection and during the infection. The cells were fixed at 24 hpi, and, after immunostaining, the infected cells were counted. The results are shown in Table 2. DES remained the most active compound, but hexestrol showed comparable activity and a larger selectivity index, while dienestrol and bakuchiol showed antiviral activity at higher concentrations. We included a positive control in the assay: namely, a functionalized cyclodextrin that has been previously shown to exert antiviral activity against influenza [13].

### 2.3. The Activity of DES Is Maintained against Multiple Laboratory and Clinical Strains

To verify whether DES activity was restricted to N09 or conserved among different strains of influenza virus, it was tested against six laboratory strains and three clinical strains isolated from patient samples collected at the University Hospital of Lausanne in 2022. DES proved to be active, with different potencies, against H1N1, H3N2, H5N1, and influenza B strains. In contrast, the compound did not inhibit an unrelated virus, SARS-CoV-2 (Table 3 and Appendix A).

### 2.4. DES Is Active in Different Cell Lines and Human-Derived Respiratory Tissues

The activity of DES was tested in parallel on MDCK, A549, Calu-3 cells, and a respiratory tissue model. The EC_50_ determined in A549 cells was comparable to that determined in MDCK, while the higher EC_50_ in Calu-3 cells might be due to a different HA protease’s activation [14] (Figure 1A). Importantly, we also tested the antiviral activity in a respiratory tissue model (Mucilair) that faithfully represents the composition of the human upper-airway epithelium. The treatment with DES (20 µM) was started 2 h post-infection to recapitulate physiological conditions with a clinical influenza A strain (H1N1). Apical washes of the tissues were collected every day. DES showed inhibitory activity, significantly reducing the viral load by more than one log up to the endpoint of the experiment at 96 hpi (Figure 1B) in the absence of toxicity (Appendix A).

### 2.5. DES Acts on an Initial Phase of Viral Infection

In order to investigate the mechanism of action of DES, time-of-addition assays in the absence of trypsin were performed to have a single replication cycle, in opposition to what is shown in Figure 1B. The compound was added at different time points from 24 h before infection to 4 h after infection (Figure 2A,B). The results evidenced good inhibition in the pre-treatment (Figure 2A), while the potency of the compound was lost when added 2 h post-infection (Figure 2B), demonstrating an early-phase inhibition. Binding experiments were performed by pre-treating the cells for 1 h at 37 °C and adding the virus for 1 h at 4 °C in the presence of the compound. At the end of the incubation, the cells were lysed, and the amount of viral RNA was measured by RT–qPCR. The results (Figure 2C) evidence that the attachment of the virus was not inhibited. The mechanism of action was then further demonstrated by an RNA kinetics evaluation (Figure 2D). Viral RNA replication in the presence of the drug was inhibited at 6 h post-infection. Therefore, inhibition is likely to occur in the entry process, since attachment is not impaired, but replication is inhibited.

### 2.6. The Target of DES Is the Viral Hemagglutinin

Finally, we tested the genetic barrier to resistance by growing the virus in the presence of increasing concentrations of DES. After the fifth passage, the virus was tested against DES and did not show any residual inhibitory activity. Therefore, all of the viral genome segments of the resistant virus and the virus passaged in parallel in the absence of the drug were sequenced. Two conserved mutations in the hemagglutinin in two independent replicates were identified. The first is a C44T (A15V) mutation in the signal peptide, while the second is T382A (F128I) in the head domain. These mutations were searched in the sequences available in the GISAID database between 2015 and 2022. C44T was found in 0.068% and T382A was found in 0.023% of the sequences, showing their rarity. Furthermore, in the apical washes of the Mucilair tissues treated with the compounds at 4 dpi (Figure 1B), the mutations were not present.

We then investigated, through in silico methods, the location of the mutations and the presence of a putative binding pocket (Figure 3C). The structure of H1 influenza from PDB 3UBQ [15] was used to visualize the locations of the mutations present in the resistant influenza. A15 was not present in the mature hemagglutinin. At the location of F128, no docking poses were identified with DES. Therefore, blind docking was performed with SwissDock to identify potential binding sites. We localized three potential binding sites in the head of the hemagglutinin and one in the stem. Two sites were approximately 10 Å away from F128. We then re-docked DES using Maestro Schrödinger using Glide XP and Prime MM-GBSA. DES interacts mainly with its phenol groups through H-bond and Pi interactions, as observed in the different binding sites, with estimated ΔG_mm-gbsa_ between −52.20 and −32.97 kcal/mol (Appendix A). According to the available structures of different subtypes of the hemagglutinin (H1, H3, H4, H5, H7, H10, and H18), the binding site in the head interacting with Ser92 and Ala278 in H1 could be conserved, supporting the antiviral activity on different subtypes of influenza virus (Table 3 and Appendix A). Furthermore, we verified that DES could bind to the influenza B hemagglutinin. We performed docking on influenza B/Yamagata (PDB 4M44 [16]) at the corresponding binding pocket of the most favorable site according to the ΔG_mm-gbsa_ seen with H1 influenza A. Although influenza B belongs to another genus, DES was predicted to bind to the same pocket with a ΔG_mm-gbsa_ of −27.28 kcal/mol, computationally confirming the antiviral activity seen in this virus (Table 3, Appendix A).

### 2.7. In Vivo Testing

To test whether the effect observed in cells and tissues was maintained in vivo, we performed a proof-of-concept antiviral assay in female BALB/c mice. The mice were treated intranasally with hexestrol without any specific formulation. We selected this compound since it showed lower toxicity compared to DES and selected a low dose to avoid any toxic effects. Mice were treated daily intranasally with 60 μg/day of hexestrol and infected with a lethal dose of N09 [17]. The weight loss was monitored daily, and the results are presented in Figure 4. In the presence of the compound, the mice exhibited significantly lower weight loss, reduced morbidity, and less severe hypothermia (Appendix A). The reduced morbidity parameters allowed 20% of treated animals to survive the lethal challenge (Figure 4). In order to compare morbidities, we calculated the area under the weight curve (AUC) for each infected mouse. Only the values obtained during the first 5 days were used since mortality occurred after this time point. The data obtained confirmed a significant effect of the hexestrol treatment (Figure 4C).

## 3. Discussion

Influenza virus is one of the most widespread respiratory infections, and the arsenal of antiviral drugs available is not sufficient. In this context, we screened a library of compounds, and we discovered the antiviral activity of DES and structurally related molecules against multiple strains of influenza virus.

DES is mostly active against H1N1 strains but retains activity against H3N2, H5N1, and influenza B (Table 3). The antiviral activity is conserved in different cell lines and in human-derived pseudostratified epithelia derived from the upper respiratory tract when added post-infection (Figure 1). From time-of-addition assays, kinetic experiments, and binding assays, we identified that the step inhibited is likely the entry since we did not observe inhibition in binding, while in the presence of DES, replication was impaired. Moreover, we were able to select for resistance, showing the appearance of mutations in the hemagglutinin in a pocket not involved in sialic acid binding but associated with the conformational change necessary for viral entry, confirming the mechanism of action.

Moreover, we verified the in vivo activity with a proof-of-concept study by administering hexestrol intranasally. We observed a significant reduction in weight loss and the survival of 20% of mice. The results could also be related to the immunomodulatory activity of estrogens in females, as previously shown with 17β estradiol and estriol [18,19]. However, the intranasal administration should have limited the systemic effects. The possibility that hexestrol may induce a protective response in the respiratory sphere via a transcriptomic action cannot be excluded. To our knowledge, the hexestrol effect on the respiratory epithelium has not been described, and such characterization should be conducted to optimize the effectiveness of this molecule. Although the activity of hexestrol in vivo is promising, further optimization is needed. Hexestrol was not formulated for intranasal administration, and we used a highly virulent mouse-adapted N09 strain that might not be recapitulative of physiological viral replication kinetics in vivo. The half-life of the compound in the airways could be a limiting factor in its effectiveness. Therefore, pharmacokinetic studies could be informative regarding the availability of the reagent in situ to inhibit the virus. As the replication cycle of the influenza virus is between 6 and 9 h, it is likely that the bioavailability of hexestrol, which is only administered every 24 h, is not sufficient to fully control the virus. Multiple daily treatments would be required, and therefore, a nebulized formulation should be considered to avoid anesthetizing the animals multiple times daily.

Overall, our work presents two main limitations: DES is a non-steroidal compound with estrogenic activity that was used for miscarriages and to treat some forms of cancer; however, the compound was found to be teratogenic and carcinogenic and therefore discontinued [20]. The carcinogenic activity was evidenced with systemic and repeated administration, while in our case, we investigated a topical route. Intranasal administration was previously investigated for other estrogens, and it leads to a shorter time in the plasma for the compound [21], which might limit the systemic side effects. Moreover, influenza is an acute respiratory infection; therefore, the treatment of influenza-infected patients is limited in time. Importantly, hexestrol, which was used for in vivo experiments, is associated with fewer side effects compared with DES, showing that it is possible to find analogs retaining potent activity against influenza virus. Therefore, further research on analogs and molecules binding to the same binding pocket can overcome this limitation.

The second limitation is related to the emergence of resistance. We observed the emergence of resistance to DES after five passages. The mutations selected in the clones resistant to DES, however, are not common mutations present in the circulating strains, which we verified through an analysis using the GISAID database, where these mutations appeared at frequencies of less than 0.1% between 2015 and 2022. Furthermore, although rapidly appearing in cell lines, in the ex vivo experiments, we did not observe the emergence of mutations throughout the 4 days of infection. 

It is important to underline that with influenza strains, escape mutants are nearly inevitable due to the high mutation rate, as demonstrated by the observation of resistance with all of the commercial antivirals available [1]. However, the problem could be overcome by combinatorial therapies, as demonstrated by previous studies [22].

To better understand the structural features leading to the hemagglutinin interaction, we tested analogs of DES. We identified hexestrol, dienestrol, and bakuchiol as influenza inhibitors. In line with our results, bakuchiol was previously shown to inhibit influenza virus with a similar profile to what was observed with DES. In the presence of the compound, the authors observed a decrease in viral RNA and viral proteins, but the activity of hemagglutinin was not impaired, according to hemagglutination and furin cleavage assays, but specific entry assays and resistance selection assays were not performed [23]. The two molecules, given their similar structural features, could therefore have the same mechanism of action. In contrast to our results, resveratrol was previously shown to inhibit influenza [24]; however, the mechanism of action was related to a rather late stage of replication that we did not investigate. In conclusion, from the analysis of the antiviral activity of DES analogs, we can conclude that the phenol is fundamental for the activity. Moreover, from the docking analysis, no binding sites were found at the location of the mutated residue, F128. However, we were able to find several potential binding sites, with two close to F128. These binding regions are of particular interest because a previous study on H5 showed antibody binding in proximity to them [25]. The neutralization mechanism of the antibody suggested the inhibition of the pH-induced conformational change in HA involved in viral entry. Like the aforementioned antibody and the compound JNJ4796 [26], DES could therefore have a fusion inhibition mechanism. Importantly, previous studies on other compounds binding to the hemagglutinin and preventing entry identified resistance appearing in proximity to the binding pockets but not in residues directly involved in it [27], in concordance with our findings. Nevertheless, further investigations are needed to validate the putative binding sites identified and to clarify the role of the mutation (F128I) in the loss of activity of the compound. From the study of the analogs and from the identification of the binding site, additional molecules can be selected from in silico screening to bind to the HA site identified, with increased potency against H3N2 and influenza B strains.

In conclusion, in the quest for new molecules active against influenza virus, the antiviral activity of DES and hexestrol deserves further investigation to identify analogs binding to the same site on the hemagglutinin with increased potency, a higher barrier to resistance, and limited activity on the estrogen receptor.

## 4. Materials and Methods

### 4.1. Cells and Viruses

Madin-Darby canine kidney (MDCK) (CCL-34-ATCC), MDCK SIAT (from the Influenza Reference Center of Switzerland), A549 (CCL-185-ATCC), Vero E6 (CRL-1586-ATCC), and Calu-3 (HTB-55-ATCC) cells were propagated in Dulbecco’s Modified Eagle’s Medium (DMEM), high glucose + Glutamax, supplemented with fetal bovine serum (FBS) 10% and penicillin/streptomycin (pen/strep) 100 UI/mL at 37 °C with 5% of CO_2_.

Influenza A virus strains H3N2 A/Wyoming/2003/3 (WYO), H1N1 A/Puerto Rico/8/34 (PR8), H1N1 A/Netherlands/602/2009 (N09), H1N1 A/WSN/1933, and H5N1 A/Viet Nam/1203/2004, were obtained from the laboratory of Prof. Mirco Schmolke and were propagated in MDCK cells at 37 °C in the presence of TPCK-treated trypsin (Sigma Aldrich, St. Louis, MO, USA) (0.4 μg/mL). B/Washington/02/2019 was obtained from the Sentinella Center for influenza virus at the University Hospital of Geneva and was propagated in MDCK cells at 33 °C. Clinical strains were isolated directly from clinical specimens of MDCK cells for H1N1 strains and of MDCK-SIAT for H3N2 strains. SARS-CoV-2 B.1.1.7 (hCoV-19/Switzerland/VD-CHUV-GEN3159/2021) was isolated and propagated in Vero E6 from a clinical sample from the University Hospital of Lausanne (CHUV), as described previously [28]. Supernatants were collected, centrifuged, and aliquoted at −80 °C.

### 4.2. Molecules

DES (46207, Sigma), dienestrol (46190), resveratrol (554325), hexestrol (H7753), dextromethorphan (PHR1018), bakuchiol (68612), and masoprocol (74540) were purchased from Sigma Aldrich (Sigma Aldrich, St. Louis, MO, USA). They were suspended in dimethyl sulfoxide (DMSO) at a concentration of 10 mM, aliquoted, and stored at −20 °C.

### 4.3. HTS Screening

A549 (4000 cells/well) were seeded in 384-well plates. The following day, 10 µM of compounds belonging to the Prestwick library (#651201) were added to cells for 1 h at 37 °C, and then the compounds and the virus (N09 MOI 0.001 or MOI 0.01) were added to cells for 48 h. At the end of the incubation, the cells infected with MOI 0.001 were fixed with methanol and immunostained with Flu A antibody (Light Diagnostic 5001, Merck, MO, USA) and a horseradish peroxidase-conjugated mouse secondary antibody (Cell Signaling Technologies, Danvers, MA, USA), and subsequently, tetramethylbenzidine (TMB) was added to the cells and blocked with 1N Chloridric Acid. The absorbance was read at 450 nm with SpectraMAx Paradigm (Molecular Devices, San Jose, CA, USA). The percentage of infection was calculated by subtracting the value of cells not infected and then dividing the absorbance of the infected and treated wells by the mean of the absorbance of infected wells. The percentage of inhibition was calculated by subtracting the value from 100. Alternatively, methyl tetrazolium bromide (MTT) was added to the cells infected with MOI 0.01 for 3 h at 37 °C, the cells were lysed with DMSO, and absorbance was read at 570 nm with SpectraMAx Paradigm (Molecular Devices, San Jose, CA, USA).

### 4.4. Hit Selection

The plates were validated by calculating the Z score (the number of standard deviations from the mean score) of each compound and considering valid only the plates with Z factor > 0.4 [29]. In every plate, favipiravir was present as a positive control, together with blanks and infected cells treated with DMSO. The hits were scored by matching the results of the ELISA and the viability assay.

### 4.5. Toxicity Assays

MDCK, Calu-3, or A549 cells (16,000 cells/well) were seeded in a 96-well plate. The following day, dilutions of the molecules were made in DMEM without FBS supplementation. DMSO was used as a control. Molecules were then added to cells and incubated at 37 °C for 24 h. MTT was added, and the plate was kept at 37 °C for 4 h. The cells were lysed with DMSO, and the absorbance was read at 570 nm.

### 4.6. Antiviral Assays

MDCK, Calu-3, or A549 cells (16,000 cells/well) were seeded in a 96-well plate. The following day, successive dilutions of the molecules, starting from 10 µM, were made in a 96-well plate, in DMEM without FBS supplementation. The cells were then pre-treated for 1 h at 37 °C with the compounds and subsequently infected with N09 MOI 0.01 in the presence of the compounds and incubated for 24 h at 37 °C. Cells were fixed with methanol and immunostained with anti-Flu A antibody (Light Diagnostic 5001, Merck, St. Louis, MO, USA) and an HRP-conjugated mouse secondary antibody (Cell Signaling Technologies, Danvers, MA, USA), and subsequently, 3,3′diaminobenzidine (DAB) solution was added to the cells for 5 min. Infected cells were counted under a light microscope as previously described [17].

For SARS-CoV-2, experiments were performed similarly to the method described in [30]. Vero E6 cells (100,000 cells) were seeded in a 24-well plate. Cells were infected with 100 PFU of SARS-CoV-2 B.1.1.7, with serial dilutions of DES starting from 10 µM. After 1 h at 37 °C, the virus was removed, and DES with the same dilutions was added with 4% Avicel GP3515 in DMEM with 2.5% FBS for 2 days at 37 °C. Cells were then fixed with 4% formaldehyde and stained with Crystal Violet. Plaques were counted manually.

### 4.7. Mechanism of Action

MDCK cells (16,000 cells/well) were seeded in a 96-well plate. The following day, cells were treated at different time points, with successive molecule dilutions starting from 10 µM in DMEM without FBS supplementation. For the pre-treatment experiments, DES was added 24, 4, or 1 h pre-infection. The virus (N09 MOI 0.01) was then added to cells and incubated for 24 h at 37 °C. For the post-treatment experiments, the virus (N09 MOI 0.01) was first added to cells, and DES was added 0, 1, or 2 h post-infection. Cells were then incubated for 24 h at 37 °C. Cells were then fixed and stained as described above.

### 4.8. Kinetics

MDCK cells (16,000 cells/well) were seeded in a 96-well plate. The following day, cells were pre-treated with DES 1 and 10 µM, 1 h before infection. After a wash, the virus (N09 MOI 0.1) and DES 1 and 10 µM were both added for 1 h. Cells were then lysed at 0, 3, or 6 h post-infection. Supernatants were collected, and RNA was extracted with the total RNA kit (Omega Biotech, Norcross, GA, USA) according to the manufacturer’s instructions. RT–qPCR was performed with the forward primer GACCRATCCTGTCACCTCTGAC, the reverse primer AGGGCATTYTGGACAAAKCGTCTA, and the probe FAM-TGCAGTCCTCGCTCACTGGGCACG-BHQ1. Then, 10 µL of the qPCR mix (specific to influenza) + 2.5 µL of RNA samples was added to a 96-well qPCR plate, which was then put in a thermocycler. Samples were first heated for 2 min at 25 °C. The next steps were 15 min at 50 °C, 2 min and 30 s at 95 °C, and finally 30 s at 60 °C; the last 2 steps were repeated for 40 cycles.

### 4.9. Binding

MDCK cells (16,000 cells/well) were seeded in a 96-well plate. The following day, cells were pre-treated with DES 1 and 10 µM, 1 h before infection. After a wash, the virus (N09 MOI 0.1) and DES 1 and 10 µM were both added for 1 h at 4 °C. Cells were then lysed 1 h post-infection, supernatants were collected, and RNA was extracted with the total RNA kit (Omega Biotech, Norcross, GA, USA) according to the manufacturer’s instructions. RT–qPCR was performed as described above.

### 4.10. Resistance

MDCK cells (400,000 cells/well) were seeded in a 6-well plate. The following day, N09 MOI 0.01 was added to cells for 1 h at 37 °C. After a wash, diethylstilbestrol and hexestrol were added with an initial concentration of 1 µM in DMEM supplemented with TPCK-trypsin. After 2 days, supernatants were collected and centrifuged for 5 min at 2000 rpm. Pellets were discarded, and supernatants were titrated. Cells were then incubated for 24 h at 37 °C. Cells were fixed and stained as previously described. Infected cells were counted under a light microscope. After the titration, a new 6-well plate was infected with previous supernatants, according to new titers. DES and hexestrol concentrations were doubled at each passage. This procedure was repeated for five passages.

### 4.11. Tissues: Toxicity Assay

MucilAir™ tissues from a pool of 14 donors were purchased from Epithelix (Plan-les-Ouates, Switzerland) and maintained in a 24-well plate with Mucilair medium at 33 °C, according to manufacturer instructions. DES 10 and 30 µM was added on the apical side. The plate was incubated at 33 °C for 24 h. After the incubation, 200 µL of the medium was added on the apical side and left for 20 min. The supernatants on the apical and basal sides were separately collected. Molecules were added again to the tissues. This process was repeated at 24, 48, 72, and 96 h. CyQuant LDH cytotoxicity assay (Thermofisher Scientific, Waltham, MA, USA) was used according to manufacturer instructions, and the absorbance was read at 490 nm.

### 4.12. Tissues: Antiviral Assay

MucilAir tissues from a pool of 14 donors were purchased from Epithelix (Plan-les-Ouates, Switzerland and were maintained in a 24-well plate with Mucilair medium at 33 °C, according to manufacturer instructions. Tissues were infected with an H1N1 clinical strain (viral copies/tissues of 1 × 10^3^) on the apical side for 2 h at 33 °C. The supernatant was removed, and the tissues were washed. DES (20 µM) was added on the apical side. The dose of the compound was selected on the basis of toxicity experiments on MucilAir. The plate was incubated at 33 °C for 24 h. After the incubation, 200 µL of the medium was added on the apical side and left for 20 min. The supernatants on the apical and basal sides were separately collected. This process was repeated at 24, 48, 72, and 96 h. Viral RNA extraction and amplification were performed as previously described.

### 4.13. Analysis of Resistant Mutation Frequency

Full hemagglutinin sequences of H1N1 influenza A pdm09 and seasonal obtained from human samples from GISAID were extracted between 2015 and 2022, resulting in 35,050 unique sequences. GISAID sequences were aligned with the hemagglutinin sequence from the resistance experiment. Nucleotides at mutated positions in the resistance experiment were extracted and counted according to their frequency. The numbering of the sequence corresponds to influenza A/California/07/2009 (NC_026433). This analysis was performed with Python 3.9 using the package BioPython.

### 4.14. In Silico Methods

PDB 3UBQ (H1 influenza) [15] was prepared with Protein Preparation Wizard (Maestro 12.8.117 Schrödinger) considering a pH of 7 and restrained minimization with the OPLS4 force field. A blind docking was performed using SwissDock (Swiss Institute of Bioinformatics, Lausanne, Switzerland) using DES as a ligand to identify potential binding sites [31]. DES prepared with LigPrep (Maestro 12.8.117 Schrödinger) was docked again with Glide XP (Maestro 12.8.117 Schrödinger) at the different sites. Prime MM-GBSA (Maestro 12.8.117 Schrödinger) was then used to refine the docking poses. The different binding sites were then visualized with MOE 2019.0102. The numbering of the interacting residues corresponds to influenza A/California/07/2009 (NC_026433).

PDB 4M44 (Influenza B) [16] was prepared similarly to HA of influenza A. After alignment with Maestro 12.8.117 Schrödinger, the corresponding interacting residues (Ser92 and Ala278) of DES on H1 influenza A from PDB 4M44 (Lys80 and Gly275, respectively) were taken as the center during grid generation. DES was docked with Glide XP and refined with Prime MM-GBSA. After visual inspection, interacting residues were visualized with MOE 2019.0102. The numbering of the interacting residues corresponds to the original PDB.

### 4.15. In Vivo Experiments

Female specific-pathogen-free BALB/c mice were purchased from the “Centre d’Elevage R. Janvier” (Le Genest Saint-Isle, France) and were used at 7 weeks of age. Mice were housed in negative-pressure isolators in a containment level 2 facility. Food and water were available ad libitum. After a stalling period of one week, the animals were infected with a lethal dose of a mouse-adapted N09, highly pathogenic in mice [17]. Mice were lightly anesthetized with a mixture of ketamine and xylazine (60 mg/kg and 12 mg/kg, respectively) and inoculated intranasally with 10 PFU of N09 in 50 μL of PBS. Two groups of animals were then formed: a group treated intranasally with a 60 µg dose of hexestrol and a control group treated with the vehicle alone (50 µL, PBS 60%, DMSO 40%). The treatment was then administered daily for 5 days under gas anesthesia: mice were placed in the chamber (XGI-8, Caliper Life Sciences, Hopkinton, MA, USA), and the anesthesia unit was turned on with a flow of 100% oxygen at a rate of 1.5–2 L min^−1^ mixed with around 4–5% (*v*/*v*) isoflurane delivered to the anesthesia chamber. The conditions of the mice were monitored daily. A humane endpoint was used during the survival study: mice were euthanized via cervical dislocation when body weights were reduced to 75% of the starting weights or when the rectal temperature was less than 32 °C.

## Figures and Tables

**Figure 1 ijms-24-15382-f001:**
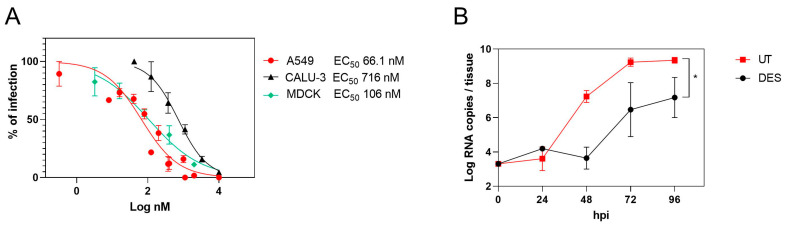
DES inhibitory activity in cells and tissues. (**A**) DES was tested in a dose–response assay in MDCK, A549, and Calu-3 cells. The percentages of infection were calculated by comparing the treated and untreated wells, and EC50 values were calculated with Prism 9. (**B**) Mucilair tissues were infected with 1 × 10^3^ RNA copies of clinical H1N1 for 2 h at 33 °C and subsequently treated with DES. Apical washes were collected every day, and RT–qPCR was performed on them. Results are the mean and SEM of 2 to 3 independent experiments. * *p* < 0.0204. DES: diethylstilbestrol.

**Figure 2 ijms-24-15382-f002:**
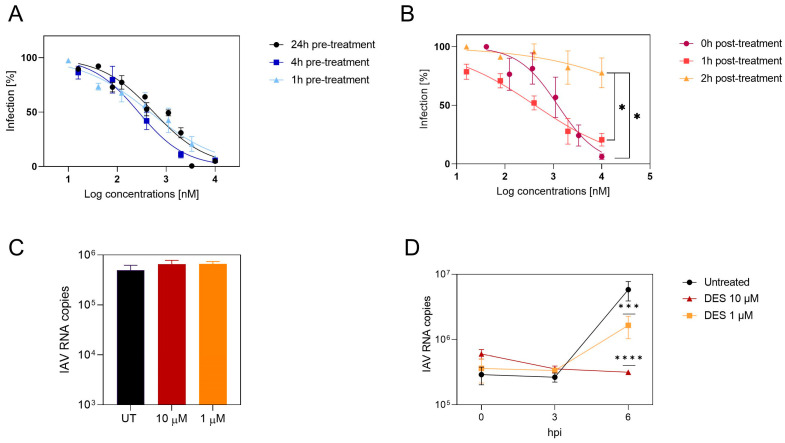
DES mechanism of action. (**A**) DES was tested in a dose–response assay in MDCK cells by adding the drug 24, 4, or 1 h before infection. (**B**) DES was tested in dose–response assay in MDCK cells by adding the drug 0 h, 1 h, or 2 h post-infection. The percentages of infection were calculated by comparing the treated and untreated wells, and EC50 values were calculated with Prism. (**C**) MDCK cells were treated for 1 h with DES at the indicated concentrations and infected for 1 h at 4 °C in the presence of the compound. The cells were then washed and lysed, and the amount of viral RNA attached was quantified by RT–qPCR. (**D**) MDCK cells were treated 1 h before infection at the indicated concentrations and during infection and lysed at 0 h, 3 h, or 6 h after infection, and RT–qPCR was performed on the lysates. Results are the mean and SEM of 2 to 3 independent experiments. * *p* < 0.0332, *** *p* < 0.0002, **** *p* < 0.0001. IAV: influenza A virus; DES: diethylstilbestrol.

**Figure 3 ijms-24-15382-f003:**
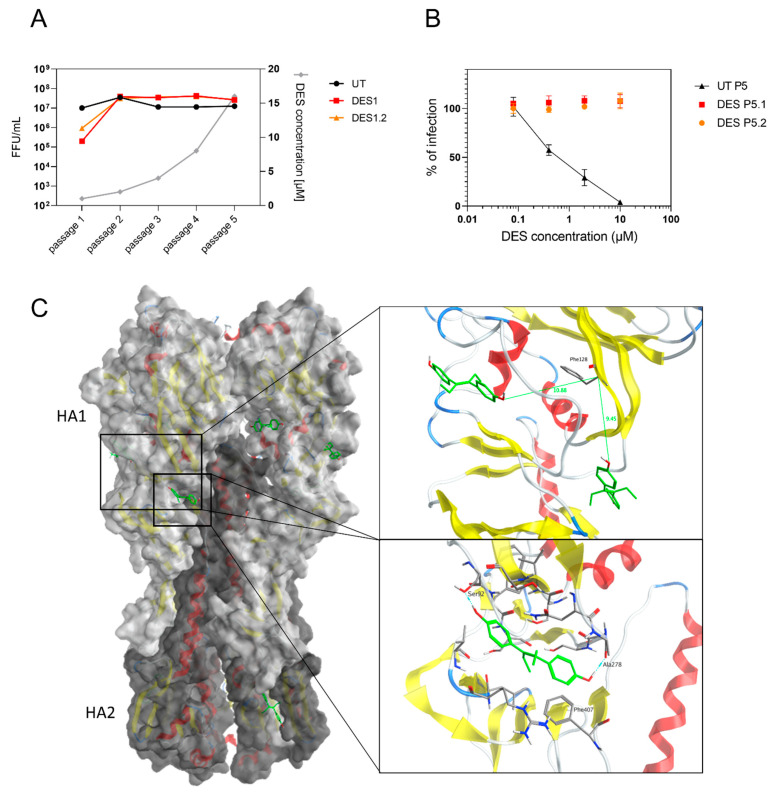
DES resistance. (**A**) N09 was passaged in the presence of increasing concentrations of DES. The supernatants were collected when the cells displayed extensive cytopathic effects and titrated. (**B**) Viruses collected at passage 5 were tested in a dose–response assay for DES inhibition. (**C**) Hemagglutinin trimer is visualized in complex with DES (green). Zoom on sites close to F128 and on one of the binding sites identified. Red color corresponds to α-helix structures, yellow color toβ-sheets structures.

**Figure 4 ijms-24-15382-f004:**
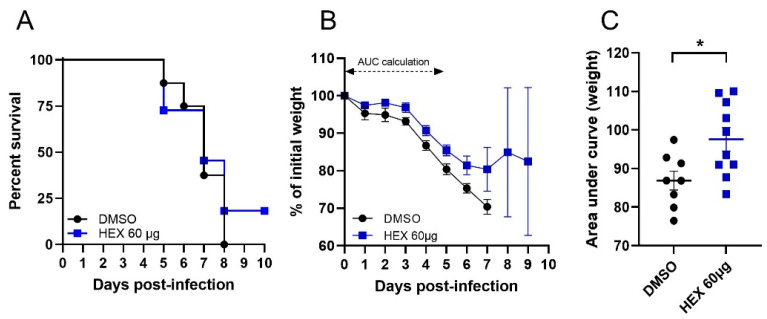
Hexestrol in vivo activity. Mice were infected with 10 PFU of N09 and treated or not with hexestrol (60 µg/mouse). (**A**) Mouse survival curves. (**B**) Daily weight monitoring of infected mice. (**C**) AUC calculation of weight curve. AUC was calculated for each individual weight curve during the 5 first days of infection. An asterisk (*) indicates a significant difference (*p* < 0.05) obtained by the Mann–Whitney test.

**Table 1 ijms-24-15382-t001:** Top hits from the Prestwick screening.

Compound	Inhibition (%)	Viability (%)
Alverine citrate salt	99	145
Dextromethorphan hydrobromide monohydrate	100	103
Camylofine chlorhydrate	97	97
Procyclidine hydrochloride	97	91
Diethylstilbestrol	114	72

**Table 2 ijms-24-15382-t002:** Antiviral activity of DES analogs.

	Structure	EC_50_ (95% CI) (µM)	CC_50_ (µM)	SI
Diethylstilbestrol	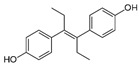	0.12 (0.06–0.17)	9.81	89.2
Dienestrol	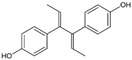	3.13 (2.09–4.78)	32.6	10.4
Resveratrol	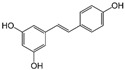	>10	52.0	n.a.
Hexestrol	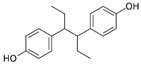	0.14 (0.05–0.25)	46.9	335
Bakuchiol	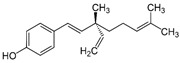	0.94 (0.65–1.35)	46.2	49.1
Masoprocol	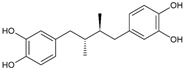	>10	84.9	n.a.
CD-MUS		13.4 (10.6–17.3)	>100	>7.46

Half-maximal effective concentration (EC_50_), Confidence Interval (CI), cytotoxic concentration (CC_50_), selective index (SI), not assessable (n.a.).

**Table 3 ijms-24-15382-t003:** Antiviral activity of DES against different influenza virus strains and subtypes.

Strains	EC_50_ (95% CI) (µM)
H1N1 A/Puerto Rico/8/34	1.64 (1.18–2.25)
H1N1 A/WSN/1933	1.11 (0.79–1.59)
H1N1 A/Netherlands/602/2009	0.12 (0.06–0.17)
H1N1 A/Lausanne/2022 Clinical 1	0.56 (0.48–0.65)
H1N1 A/Lausanne/2022 Clinical 2	0.37 (0.28–0.48)
H3N2 A/Wyoming/2003/3	3.81 (2.73–5.16)
H3N2 A/Lausanne/2022 Clinical 3	4.71 (3.27–6.83)
H5N1 A/Viet Nam/1203/2004	3.84 (2.64–5.89)
B/Washington/02/2019	4.15 (1.84–17.1)
SARS-CoV-2	n.a.

Half-maximal effective concentration (EC_50_), Confidence Interval (CI), not assessable (n.a.).

## Data Availability

Raw data are available from the authors upon reasonable request.

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
