# Peer review of "Non-Steroidal Estrogens Inhibit Influenza Virus by Interacting with Hemagglutinin and Preventing Viral Fusion"

_ijms, 2023, doi:10.3390/ijms242015382_

Round 1
Reviewer 1 Report
The work that was done by Franzi et al. is devoted to the repurposing of FDA-approved drugs for targeting the Influenza virus. Although the effect of DES on the virus seems to be valid and genuine, the concerns regarding the teratogenicity and carcinogenicity of DES seem to be very significant limitations for the approval of the drug for the treatment of these seasonal infections. Apart from this, the localization of the binding pocket was done only computationally and was purely based on the docking results - no experimental validation or even MD of this data has been provided here. However, considering that such limitations have been explicitly stated in the paper, I think the work may be published in the IJMS as is.
Additional comments
I would like to extend my comment regarding the ijms-2665367 manuscript
The manuscript authored by Franzi et al. appears to me as a good example of the research about the repurposing of drugs. I have no doubts regarding the novelty of the particular research, although the methods that were used there are already well-established; nevertheless, the results that were presented to the best of my knowledge are original and novel. I believe this manuscript may be of interest to the readers of IJMS as it evaluate the developed candidate from both the experimental and computational point of view. As I have mentioned in my review, there are some particular limitations in the research considering the potential applications of the identified candidates; however, the authors explicitly mentioned these limitations in manuscript. It is hard to predict if these limitations are serious or not, but the manuscript is still of interest to the audience of IJMS from my point of view.
Author Response
The work that was done by Franzi et al. is devoted to the repurposing of FDA-approved drugs for targeting the Influenza virus. Although the effect of DES on the virus seems to be valid and genuine, the concerns regarding the teratogenicity and carcinogenicity of DES seem to be very significant limitations for the approval of the drug for the treatment of these seasonal infections. Apart from this, the localization of the binding pocket was done only computationally and was purely based on the docking results - no experimental validation or even MD of this data has been provided here. However, considering that such limitations have been explicitly stated in the paper, I think the work may be published in the IJMS as is.
-We thank the reviewer for her/his comments. We agree with the reviewer about the limitations of our work, we mentioned originally in the text the limitations of DES as treatment, and we have now added a sentence in the discussion about the further experiments needed to validate the localization of the binding pocket:
“We were able to find several potential binding sites with two close to F128. These binding regions are of particular interest because a previous study on H5 showed antibody binding in proximity [25]. The neutralization mechanism of the antibody suggested an inhibition of the pH-induced conformational change of HA involved in the viral entry. Like the aforementioned antibody and the compound JNJ4796 [26], DES could therefore have a fusion inhibition mechanism. Importantly, previous studies on other compounds binding the hemagglutinin and preventing entry identified resistance appearing in proximity to the binding pockets but not in residues directly involved in it [27], in concordance with our findings. Nevertheless, further investigations are needed to validate the putative binding sites identified and to clarify the role of the mutation (F128I) in the loss of activity of the compound.”
Additional comments
I would like to extend my comment regarding the ijms-2665367 manuscript
The manuscript authored by Franzi et al. appears to me as a good example of the research about the repurposing of drugs. I have no doubts regarding the novelty of the particular research, although the methods that were used there are already well-established; nevertheless, the results that were presented to the best of my knowledge are original and novel. I believe this manuscript may be of interest to the readers of IJMS as it evaluate the developed candidate from both the experimental and computational point of view. As I have mentioned in my review, there are some particular limitations in the research considering the potential applications of the identified candidates; however, the authors explicitly mentioned these limitations in manuscript. It is hard to predict if these limitations are serious or not, but the manuscript is still of interest to the audience of IJMS from my point of view.
-We thank the reviewer for his encouraging feedback. We are aware of the limitations that represent DES for a treatment in patients. However, we think that our research could help to find molecules with similar mechanisms of action devoid of side effects.
Reviewer 2 Report
The authors of this paper were on a mission to discover a new class of antivirals against a broad spectrum of influenza viruses. They identified diethylstilbestrol (DES) and hexestrol as two potential drug candidates. The following in vitro and vivo assays were conducted to investigate the administration window, drug target, and mechanism. Most of the assays were rationally designed and provided positive results. The drug target (hemagglutinin) was identified through multiple infection cycles and spotted the mutated viral protein genes from the resistant viral strain. However, the mechanism study of DES acts as a viral entry inhibitor is not compelling (section 2.5). Figure 2D shows that viral RNA replications are mostly unimpaired regardless of DES concentration, which makes the conclusion, “but the replication is inhibited (lines 127-128)” confusing. Figure 2C shows between 0-3 hours post-infection, DES-loaded cells have slightly higher viral RNA copies than untreated cells. Not only does it contradict Figure 2D, but the results are also puzzling since it may suggest at an early stage, the DES may promote viral replication instead of inhibition. I suggest section 2.5 be clarified before the paper is accepted for publication.
Author Response
The authors of this paper were on a mission to discover a new class of antivirals against a broad spectrum of influenza viruses. They identified diethylstilbestrol (DES) and hexestrol as two potential drug candidates. The following in vitro and vivo assays were conducted to investigate the administration window, drug target, and mechanism. Most of the assays were rationally designed and provided positive results. The drug target (hemagglutinin) was identified through multiple infection cycles and spotted the mutated viral protein genes from the resistant viral strain. However, the mechanism study of DES acts as a viral entry inhibitor is not compelling (section 2.5). Figure 2D shows that viral RNA replications are mostly unimpaired regardless of DES concentration, which makes the conclusion, “but the replication is inhibited (lines 127-128)” confusing. Figure 2C shows between 0-3 hours post-infection, DES-loaded cells have slightly higher viral RNA copies than untreated cells. Not only does it contradict Figure 2D, but the results are also puzzling since it may suggest at an early stage, the DES may promote viral replication instead of inhibition. I suggest section 2.5 be clarified before the paper is accepted for publication.
We thank the reviewer for his comment. We changed the order of the figure by switching figure 2C and figure 2D to first discuss in the text about the attachment and then the replication of the virus. For Figure 2C (previously 2D), we did a binding experiment to exclude any effect (positive or negative) on the attachment of influenza on cells, and we do not see any inhibition, therefore the step inhibited is after the binding. Regarding Figure 2D (previously 2C), we performed a two-way Anova. The statistical results showed that at 0h and 3h, there are no significant difference between each group, since we do not observe yet viral replication. However, at 6h post infection, the two dose of DES (10µM and 1µM) showed significant reduction demonstrating an inhibition of viral RNA replication. We added the statistics on the graph, and we clarified in the text that from that point there is an inhibition of replication. Here below you will find the revised section 2.5 and the revised figure 2:
“In order to investigate the mechanism of action of DES we performed time of addition assays with a single replication cycle, in opposition to what shown in Figure 1B, by adding medium without trypsin post-infection. We added the compound from 24 hours before infection to 4 h after infection (Figure 2A-B). The results evidenced a good inhibition in pre-treatment (Figure 2A) while the potency of the compound was lost when added 2h post-infection (Figure 2B) demonstrating an early phase inhibition. We then performed binding experiments. We pre-treated the cells 1h at 37°C and then we added the virus for 1h at 4°C in presence of the compound, at the end of the incubation the cells were lysed, and the amount of viral RNA was measured by RT-qPCR. The results (Figure 2C) evidence that the attachment of the virus is not inhibited. The mechanism of action was then further demonstrated by RNA kinetics evaluation (Figure 2D). We observed an inhibition of the viral RNA replication in the presence of the drug at 6h post-infection. Therefore, the inhibition is likely to be in the entry process, since the attachment is not impaired, but the replication is inhibited.
Figure 2. DES mechanism of action. A) DES was tested in dose response in MDCK cells by adding the drug 24, 4 or 1h before infection. B) DES was tested in dose response in MDCK cells by adding the drug 0h, 1h or 2h post-infection. The percentages of infection were calculated comparing the treated and untreated wells and EC50 values were calculated with Prism C) MDCK cells were treated for 1h with DES at indicated concentration and infected for 1h at 4°C in presence of the compound. The cells were then washed and lysed and the amount of viral RNA attached quantified by RT-qPCR. D) MDCK cells were treated 1h before infection at indicated concentrations and during infection and lysed at 0h, 3h or 6h after infection and RT-qPCR was performed on the ly-sates. Results are mean and SEM of 2 to 3 independent experiments. * p < 0.0332, *** p < 0.0002, **** p < 0.0001 IAV: influenza A virus DES: diethylstilbestrol”
We changed a paragraph as well in the discussion to better clarify the mechanism of action
“From time of addition assays, kinetics experiments and binding assays, we identified that the step inhibited is likely the entry since we did not observe inhibition in binding, while in presence of DES the replication was impaired. Moreover we were able to select resistance showing the appearance of mutations in the hemagglutinin, in a pocket not involved in sialic acid binding but associated to the conformational change necessary for the viral entry, confirming the mechanism of action.”